# Activated Carbon with Ultrahigh Specific Surface Derived from Bamboo Shoot Shell through K_2_FeO_4_ Oxidative Pyrolysis for Adsorption of Methylene Blue

**DOI:** 10.3390/molecules28083410

**Published:** 2023-04-12

**Authors:** Yuyu He, Liangmeng Ni, Qi Gao, Hao Ren, Mengfu Su, Yanmei Hou, Zhijia Liu

**Affiliations:** 1International Centre for Bamboo and Rattan, Beijing 100102, China; 2Key Laboratory of NFGA/Beijing for Bamboo & Rattan Science and Technology, Beijing 100102, China

**Keywords:** bamboo shoot shell, activated carbon, adsorption process, methylene blue, K_2_FeO_4_

## Abstract

To effectively remove methylene blue (MB) from dye wastewater, a novel activated carbon (BAC) was manufactured through co-pyrolysis of bamboo shoot shell and K_2_FeO_4_. The activation process was optimized to a temperature of 750 °C and an activation time of 90 min based on its excellent adsorption capacity of 560.94 mg/g with a yield of 10.03%. The physicochemical and adsorption properties of BACs were investigated. The BAC had an ultrahigh specific surface area of 2327.7 cm^2^/g and abundant active functional groups. The adsorption mechanisms included chemisorption and physisorption. The Freundlich model could be used to describe the isothermal adsorption of MB. The kinetics confirmed that the adsorption of MB belonged to the pseudo-second-order model. Intra-particle diffusion was the main rate-limiting step. The thermodynamic study showed that the adsorption process was endothermic and temperature was beneficial for the improvement of adsorption property. Furthermore, the removal rate of MB was 63.5% after three cycles. The BAC will have great potential for commercial development for purifying dye wastewater.

## 1. Introduction

Water pollution has a destructive and disastrous influence on the ecological environment and human health. Dye wastewater is a major type of water pollution [1], which can block sunlight penetration and inhibit photosynthesis in aquatic life [2]. Most dyes in wastewater contain non-biodegradable aromatic compounds. They accumulate in the human body through the food chain to cause carcinogenesis, mutagenicity, and teratogenesis [3]. Therefore, the effective technologies for dye removal have increasingly attracted attention, such as adsorption [4], advanced oxidation processes [5], membrane filtration [6], coagulation [7], and biological treatment [8]. Adsorption is especially recognized as a green and cheap technology due to its high efficiency, easy-operability, and cost-effectiveness.

Activated carbon (AC) has an ultrahigh specific surface area (SSA), excellent pore structure, and high adsorption activity, which is why it is widely used as a dye adsorbent [9]. It has great development potential because it can be manufactured by various abundant and low-cost biomass feedstocks, such as rice husk [10], wheat stalk [11], core straw [12], pomelo peel [13], coconut shell [14], etc. Bamboo shoot shell (BSS) is the inedible outer epidermis of bamboo shoots with a high annual production [15]. It is an inexpensive feedstock for the preparation of AC. It especially has a low degree of lignification, and the vascular bundle cells have thin walls and large cavities. These characteristics are helpful for the formation of an AC with a narrow distribution of pore size [16]. Furthermore, BBS has a large number of amino acids, which can be used as a high-quality N source for the synthesis of N-doping AC [17]. However, these compositions result in a serious waste, causing environmental pollution, if BSS is discarded. Therefore, it is very significant to utilize BSS to manufacture AC.

Inserting different elements into the surface of AC can improve the specific surface area and functional groups, especially in the case of metals [11]. Potassium ferrate (K_2_FeO_4_) is a new chemical modifier [18]. It introduces lots of oxygen-containing functional groups and γ-Fe_2_O_3_ to AC, which promotes the adsorption properties of metal ions [13]. KOH produced by K_2_FeO_4_ during the activation process corrodes the surface of AC, which further increases the adsorption characteristics of AC [19]. Furthermore, K_2_FeO_4_ introduces magnetic iron-containing material to aid in the separation of AC material and water. However, the literature on the application of AC modified by K_2_FeO_4_ mainly focuses on removing metal ions and antibiotics from wastewater [20]. There are few reports on the preparation of AC from bamboo shoot shells activated by K_2_FeO_4_ for dye removal.

The objective of this study was to obtain optimum conditions to prepare BAC for MB removal through K_2_FeO_4_ oxidative pyrolysis. Process parameters including activation temperature and activation time were optimized according to the adsorption capacity of MB. The physicochemical properties of ACs derived from bamboo shoot shells (BACs) were characterized. The factors affecting the adsorption performance of BACs were investigated, including temperature, pH, adsorption time, and concentration of MB. The adsorption mechanisms of BACs were analyzed through the fitted isotherm, kinetic models, and thermodynamics study.

## 2. Results and Discussion

### 2.1. Effect of Activation Parameters on Preparation of BACs

Figure 1 shows the MB adsorption capacity of BACs. When the activation time remained unchanged at 120 min, activation temperatures had a significant influence on MB adsorption of BACs. When the activation temperature increased from 600 °C to 750 °C, the adsorption capacity of BACs gradually increased. The BACs with an activation temperature of 750 °C had the maximum adsorption capacity (476.3 mg/g). K_2_FeO_4_ reacted with C during the activation process, which are shown in Equations (1)–(9) [11]. KOH produced by K_2_FeO_4_ and H_2_O during impregnation reacted with C to form pores. Furthermore, K_2_CO_3_ and K_2_O produced by the reaction continued to react with C to improve pores [21]. This enhanced the adsorption capacity of MB. When the activation temperature was 800 °C, the MB adsorption capacity of BACs decreased because the pores of BACs collapsed. Figure 1b shows that the adsorption capacity of BACs gradually decreased when the activation time increased at an activation temperature of 750 °C. At the early stages of the activation process, the BACs produced a large number of micropores with K_2_FeO_4_. With the extension of activation time, the micropores and mesopores further expanded into large pores. When the activation time was 120 min, the MB adsorption capacity of BACs decreased. Therefore, the optimum progress of BACs with the yield of 10.03% was suggested to be at an activation temperature of 750 °C and activation time of 90 min.
K_2_FeO_4_ + 10H_2_O = 8KOH + 4Fe(OH)_3_
(1)
6KOH + 2C = 2K + 3H_2_ + 2K_2_CO_3_
(2)
K_2_CO_3_ = KO_2_ + CO_2_
(3)
CO_2_ + C = 2CO (4)
K_2_CO_3_ + 2C = 2K + 3CO (5)
K_2_O + C = 2K + CO (6)
Fe(OH)_3_ = FeO(OH) + Fe_2_O_3_
(7)
3Fe_2_O_3_ + (H_2_, C, CO) = 2Fe_3_O_4_ + (H_2_O, CO, CO_2_) (8)
Fe_3_O_4_ + 4(H_2_, C, CO) = 3Fe + 4(H_2_O, CO, CO_2_) (9)

### 2.2. Characterizations of BACs

The abundant adsorption sites of AC depended on the high SSA and pore volume, and mesopore acts as the adsorption site for MB [22]. Figure 2a shows the N_2_ adsorption–desorption isotherms of BACs prepared at activation temperatures of 600–800 °C for an activation time of 120 min. The curves of BACs prepared at temperatures lower than 700 °C were typical type I isotherms, indicating that BACs showed a narrow distribution of pore size and micropore structure [16]. When the activation temperature was 700 °C, the curve of BACs showed an obvious slope under high pressure and the emergence of hysteresis rings. This confirmed that the number of mesopores gradually increased. Figure 2b shows the pore size distribution of BACs prepared at activation temperatures of 600–800 °C. With the increase in activation temperature, the reaction became increasingly intense. This decreased the number of micropores and improved mesoporous development [23]. When activation temperature was 800 °C, the pore size of BACs further expanded due to further oxidation of C during the activation process. This resulted in the structural collapse of BACs [24]. Figure 2c shows the nitrogen adsorption–desorption isotherm of BACs corresponding to 60–180 min of activation time at 750 °C of activation temperature. There were hysteresis rings in all isotherms, indicating that mesopores improved the porosity of BACs. Figure 2d shows that the pore size of BACs was enhanced when the activation time increased from 60 min to 180 min. This confirmed that K_2_FeO_4_ further catalyzed and oxidated BACs.

Table 1 shows the SSAs and porosities of BACs. The SSA of BACs significantly depended on activation temperatures [25]. When the activation temperature was 750 °C, the SSA of BACs was the maximum at 1835.1 cm^2^/g. After K_2_FeO_4_ removed the tar and unorganized carbon at low temperatures, it promoted the formation of new pores [26]. A similar trend was observed in the total pore volume and micropore volume. However, the percentage of micropore volume decreased to 64.23% at an activation temperature of 750 °C. The maximum SSA of BACs was 2327.7 cm^2^/g, which was found at an activation temperature of 750 °C for an activation time of 90 min. The SSA and micropore ratio slightly decreased with the increase in activation time because the micropores were further enlarged during the activation process [27]. The BACs were more mesoporous and microporous when the pores were further widened with the increase in activation time. Even though mesopores were beneficial to remove dye from wastewater, the longer activation time enhanced the production cost of BACs and caused pore collapse.

### 2.3. Characterizations of Optimized BAC

Figure 3a shows that the BAC has a rough surface with some particles. Figure 3b,c confirms that the surface of BAC has a large number of pores with different pore sizes. KOH from the activation process of K_2_FeO_4_ corroded the BAC to significantly increase irregular pores [11]. This resulted in BAC having an ultrahigh SSA of 2327.7 cm^2^/g [28]. Figure 3d indicates that these particles were identified as iron species through EDS analysis. The presence of surface oxygen and nitrogen was related to the oxidation by K_2_FeO_4_ and nitrogen compounds in bamboo shoot shells, shown in Figure 3e,f [16]. These groups provided anionic adsorption sites on the surface of the BAC, which improved its affinity for MB [23].

To further investigate the form of iron species, XRD was used to determine the crystal compositions of the BAC. Figure 4a shows that the main diffraction peaks were consistent with the reference cards PDF#19-0629 (Fe_3_O_4_) [11], PDF#52-1449 (Fe_2_O_3_), and PDF#35-0772 (Fe_3_C). The peak of 2θ at 35.8° in the BAC corresponded to the diffraction of the (110) plane of Fe_2_O_3_ [29]. The four peaks at 30.1°, 43.1°, 57.6°, and 62.4° were related to Fe_3_O_4_ [30]. The Fe_3_C phase (020) corresponded to the characteristic peak at 24.4° [31]. The occurrences Fe_3_O_4_ and Fe_2_O_3_ were due to the decomposition of Fe(OH)_3_ and its reduction by C and some other reducing agents (H_2_, CO) at high temperatures [11]. In addition, the presence of iron changed the activation process of KOH, resulting in larger mesopores in the BAC [32].

Figure 4b shows the full spectrum of XPS. The C1s, N1s, O1s, and Fe2p peaks of the BAC were found at around 284.8 eV, 397.8 eV, 709.3 eV, and 531.4 eV, respectively, corresponding to the atoms accounting for 66.04%, 20.01%, 9.28%, and 4.67%, respectively. The Fe2p peak was observed in the BAC due to the formation of iron oxide, shown in Equations (7)–(9) [33]. Furthermore, the amino acids were abundant in BBS to manufacture the N-doping BAC [17]. The nitrogen groups enhanced the affinity of the BAC to water, which was conducive to the removal of pollutants in water.

Figure 4c shows that the four peaks at 284.8 eV, 285.8 eV, 288.9 eV, and 291.7 eV represented C1s, corresponding to C-C (40.66%), C-O (34.57%), C=O (13.42%), and O-C=O (11.35%), respectively [33]. The O1s spectra included four characteristic peaks. The peaks at 531.51 eV, 532.97 eV, and 534.49 eV corresponded to C=O (32.30%), C-O (31.77%), and O-C=O (11.76%), respectively [30]. The peak at 530.32 eV corresponding to Fe-O (24.17%) is shown in Figure 4d, confirming the load of Fe_x_O_y_ on the BAC [33]. The N1s spectra in Figure 4e includes three characteristic peaks at 398.01 eV of pyridinic-N, 399.96 eV of pyrrolic-N, and 402.24 eV of graphitic-N [34]. The Pyrrolic-N was the main form of nitrogen in the BAC (>atom 30%), which provided a lot of adsorption sites for MB. Figure 4f shows Fe2p3/2 and Fe2p1/2 of spin orbit splitting [33]. The photoelectron peaks at 708.75 eV and 721.57 eV corresponded to Fe^0^ [35]. The photoelectron peaks at 711.14 eV and 724.37 eV corresponded to Fe2p3/2 and Fe2p1/2 of Fe^3+^ [36]. The two binding energies at 712.67 eV and 725.39 eV were attributed to Fe2p3/2 and Fe2p1/2 of Fe^2+^ [37]. Combined with the O1s spectra, the high iron was from the decomposition and reduction of Fe(OH)_3_ into ferric oxide during the activation process. Zero-valent iron was not found in the XRD pattern because it was covered by iron oxide.

Figure 5 shows the FTIR analysis of the BAC to investigate the adsorption mechanism of MB. The adsorption peak at 3405 cm^−1^ corresponded to O-H stretching vibration [38]. The intensity of this peak decreased after the adsorption of MB, because a hydrogen bond was formed between -OH of the BAC and N groups of MB [29]. The double peaks at 2364 cm^−1^ were due to the C=O vibration of CO_2_. Compared with the adsorption peaks of the BAC, three fine peaks of high intensity were found at 2829 cm^−1^, 1596 cm^−1^, and 775 cm^−1^. They were related to C-Har bond, C=N bond and C-H bond, respectively, confirming that MB was absorbed on the BAC surface [23]. The N, S-heteroaromatic rings on MB were regarded as π-electron acceptors, which formed π–π conjugated bonds with the -OH of the BAC [39]. The adsorption band at 1363 cm^−1^ corresponded to S-O, indicating the interaction between MB and oxygen-containing groups on the surface of BAC [23]. A new absorption peak at 2082 cm^−1^ corresponded to C≡C stretching vibration because K_2_FeO_4_ oxidized the BAC. The absorption band at 562 cm^−1^ indicated the tensile vibration of Fe-O [30]. This confirmed that Fe_x_O_y_ was loaded onto the BAC [40].

Figure 6 shows the thermal stability of the BAC. The BAC had more than 80% mass yield at 800 °C of temperature. This indicated that the BAC had excellent thermal stability. The mass loss of BAC and BAC-MB at low temperatures (30–150 °C) was attributed to evaporation of water, which was consistent with the endothermic peak of DTG [23]. When the temperature was lower than 678 °C, BAC-MB had a greater mass loss. This confirmed that MB was adsorbed on the surface of the BAC. The DTG peak indicated that the process was endothermic at the pyrolysis temperature of 200–250 °C due to the breaking of chemical bonds, which were formed by the functional groups on the surface of MB and BAC.

### 2.4. Adsorption Properties of Optimized BAC

Figure 7 shows the adsorption properties of optimized BAC at different initial solution pH values. It indicated that the acidity and basicity of the dye solution can affect the adsorption property of BAC. The maximum adsorption capacity of the BAC (560.94 mg/g) was found at the pH value of 7 because of MB having a positive charge. When the pH value was lower than 7, a lot of H^+^ ions in the solution competed and occupied the adsorption site of MB to decrease the removal efficiency of the BAC [11]. When the pH value was higher than 7, the BAC was positively charged, which led to electrostatic repulsion between the BAC and MB to decrease MB removal [16]. Therefore, the pH value of the dye solution was suggested to be 7 during the adsorption process.

Langmuir, Freundlich, Temkin, and Dubinin-Radushkevich isotherm models were used to evaluate the adsorption capacity and mechanism [29]. The equations are expressed as follows:(10)$$\frac{q_{e}}{C_{e}}\;=\frac{1}{K_{L}q_{m}}\;+\frac{C_{e}}{q_{m}}$$
(11)$$R_{L}=11+C0$$
*q_e_* = *K_F_Ce*^1/n^
(12)
*q_e_* = *BlnK_T_* + *BlnCe*
(13)
*lnq_e_* = *lnq_m_* − *kε*^2^
(14)
*ε* = *RTln*(1 + 1/*Ce*) (15)
(16)$$E=1/\sqrt{2k}\;$$
where *q_e_* (mg/g) is the adsorption capacity of the BAC at equilibrium, *Ce* is the equilibrium concentrations of MB (mg/L), *q_m_* is the maximum adsorption capacity (mg/g), *K_L_* is the Langmuir constant related to the rate of adsorption (L/mg). *R_L_* value illustrates whether the adsorption process was unfavorable (*R_L_* > 1) or favorable (0 < *R_L_* <1). *K_F_* is the Freundlich constant and 1/n is the heterogeneity factor. *K_T_* is the equilibrium binding constant. R is the general gas constant (8.314 J/mol/K) and T is the adsorption temperature (K).

Figure 8 shows the fitting lines of the different models, and Table 2 lists the isotherm parameters of the BAC for MB adsorption. The R^2^ value of the Langmuir isotherm was 0.9414. This indicated that the adsorption of the BAC was monolayer adsorption [41]. Furthermore, the adsorption capacity of the BAC was 745.23 mg/g, which was calculated by the Langmuir model. The *R_L_* values gradually decreased from 7.63 × 10^−3^ to 2.56 × 10^−3^ when the concentrations of MB increased, indicating that the higher concentration was helpful for MB adsorption of the BAC [16]. Further, the *R_L_* values for BAC under all concentrations of MB were close to zero, indicating a very favorable adsorption of the MB substrate [42]. Freundlich isotherm models also had a high R^2^ of 0.9664. This confirmed that the MB adsorption of the BAC was a multi-layer physisorption process and the surface of the BAC was heterogeneous, which was consistent with SEM results [41]. The 1/n value was 0.072, which indicated that the MB adsorption of the BAC was facile and favorable [43]. Therefore, it was concluded that physisorption and chemisorption simultaneously occurred during the MB adsorption process of the BAC [44]. Furthermore, the R^2^ value of 0.9511 from the Temkin model indicated that chemisorption was also an important adsorption process [41]. The free energy of adsorption (*E*) at 1.656 kJ/mol confirmed that the main adsorption process of the BAC was physisorption [45]. In conclusion, the adsorption mechanisms of BAC were dominated by both physisorption and chemisorption [41].

Adsorption kinetics of the BAC were fitted by pseudo-first-order, pseudo-second-order, and intragranular diffusion models (see Figure 9). The models are shown in Equations (17)–(19). As shown in Table 3, the pseudo-second-order model has a higher R^2^ value of 0.9455, indicating that it was more consistent with the adsorption process of MB [22]. Furthermore, the adsorption capacity of the BAC calculated by the pseudo-second-order model was similar to the experimental adsorption capacity, indicating the existence of chemisorption during the MB adsorption process of the BAC. Chemisorption increased the affinity of the BAC for MB adsorption and reduced its release [29]. Based on the intra-particle diffusion model, the adsorption process of the BAC included three stages. The first stage occurred at 5–10 min. A lot of binding sites of the BAC rapidly transferred MB molecules from the bulk solution [46]. The MB diffused to the inner pore from the surface of the BAC at the second stage at 10–60 min. The adsorption reached equilibrium at the third stage. Figure 9c confirms that all linear extension lines of the internal diffusion model did not pass through the origin at three stages. This indicated that the liquid film or the internal diffusion particles were not the only rate-limiting steps for the BAC [16]. *C*_1_ and *C*_2_ reflected the liquid film diffusion and intra-particle diffusion, respectively [47]. The higher *C*_2_ value indicated that internal diffusion controlled the BAC adsorption rate. The ultrahigh SSA and abundant internal pores of the BAC improved MB to reach interior sites [44]. The higher initial concentration of MB increased the values of both *C*_1_ and *C*_2_, indicating that the liquid film diffusion and intra-particle diffusion were the main adsorption processes [48].
(17)$$\frac{1}{q_{t}}=\;\frac{1}{{k1q}_{e}t}+\;\frac{1}{q_{e}}$$
(18)$$\frac{t}{q_{t}}\;=\;\frac{1}{k_{2}q_{e}^{2}}\;+\;\frac{t}{q_{e}}$$
*q_t_* = *k_i_t*^1/2^ + *C*
(19)
where *q_t_* (mg/g) and *q_e_* (mg/g) are the MB adsorption capacities at various and equilibrium times, respectively. *k*_1_ (min^−1^), *k*_2_ (g/mg/min) and *K_i_* (g/mg/min^1/2^) are the constants of pseudo-first-order, pseudo-second-order and intraparticle diffusion model, respectively. *C* reflects the thickness of the boundary layer and the higher *C* value indicates its greater influence on the adsorption rate.

To analyze the isothermal data at different temperatures, *K_L_* calculated by the Langmuir model was plotted against 1/*T*. The Δ*H*° and Δ*S*° were calculated by the slope of the line and the intercept according to Equation (19). The Δ*G*° was obtained by the van’t Hoff Equation (20).
(20)$${lnK}_{L}=\frac{\Delta S{}^{\circ}}{R}\;-\frac{\Delta H{}^{\circ}}{RT}$$
Δ*G*° = −*RTlnK_L_*
(21)

Table 4 shows the values of adsorption thermodynamics. The value of Δ*G*° was negative, indicating that the adsorption of MB on BAC was a spontaneous process [42]. Furthermore, the values of ΔG° decreased when the temperature increased from 288 to 318 K. This confirmed that the temperature was conducive to the spontaneous adsorption of MB on the BAC [49]. The value of Δ*H*° was positive, indicating that the adsorption process was endothermic and temperature enhanced the adsorption property of the BAC [49]. The adsorption process of MB endothermically improved the disorder of the adsorption system. Thermodynamic parameters indicated that the temperature was beneficial to the removal of MB on the BAC.

According to the isothermal model, the adsorption behavior of the BAC depended on the combined action of physisorption and chemisorption. Figure 10 shows the adsorption mechanism of the BAC. Physisorption mainly included hydrogen bonding and pore filling. It was dominated by pore filling due to the distribution of pore size and the ultrahigh SSA of the BAC. The mesopores of the BAC promoted the adsorption process because they reduced steric hindrance. According to FTIR analysis, chemisorption involved complexation of functional groups and π-π interaction between the BAC and MB [29]. Both liquid diffusion and intraparticle diffusion controlled the adsorption process of MB. Furthermore, the value of Δ*H*° was positive. This confirmed that MB adsorption of the BAC was an endothermic process.

In order to evaluate the reusability of the BAC, it was regenerated by ethanol in this study. Figure 11 shows that the removal rate of BAC on MB decreased with the increase in the cycles. This indicated that ethanol could not completely remove MB from the surface of the BAC due to the stable chemical bonds between MB and the BAC. However, the removal rate of MB was 63.5% when the BAC was reused in three cycles. This indicated that BAC had a good adsorption performance.

## 3. Materials and Methods

### 3.1. Materials

The raw material of the bamboo shoot shell (*Phyllostachys edulis*) was collected from Fujian Province, China at 118.32° E, 27.05° N. After washing and drying, the BSS was pulverized to obtain particles with a diameter of 0.18 to 0.25 mm. Methylene blue (MB) was purchased from Rhawn. The molar mass of MB is 319.85 g/mol. The solubility of MB is 50 g/L. K_2_FeO_4_ was provided through Aladdin, Shanghai, China.

### 3.2. Preparation and Characterization of BACs

The BBS was pyrolyzed at 450 °C of temperature in a tubular furnace and was designated as BC-450. A total of 5 g of BC-450, 12.5 g of K_2_FeO_4_, and 250 mL of deionized water were mixed in a beaker. The blends were stirred for 12 h and dried in an oven at 105 °C of temperature to evaporate the water. They were pyrolyzed at an activation temperature of 750 °C for 1.5 h of activation time in a tubular furnace under an N_2_ atmosphere. The obtained samples were washed using HCl of 1 mol/L and deionized water to remove superfluous activators until the pH value of the washing solution was 7.0. Finally, they were dried at 105 °C of temperature for 24 h and were resigned as BAC, where B stands for biomass, A stands for activation, and C stands for carbon.

The morphology and structure of BACs were characterized using a scanning electron microscope (SEM-EDS, GeminiSEM360, Zeiss, Oberkoche, Germany) and an X-ray powder diffractometer (XRD, X PERTPRO-30X, Philips, Amsterdam, The Netherlands) (Appendix A). The chemical properties on the surface of BACs were investigated by an X-ray photoelectron spectrometer (XPS, Thermo, Thermo Fisher, Waltham, MA, USA) and a Fourier transform infrared spectrometer (FTIR, Nexus 670, Thermo Electron, Waltham, USA). N_2_ adsorption/desorption isotherms were acquired to evaluate the pore structure of BACs. The Brunauer–Emmett–Teller (BET) method and non-local density functional theory (NLDFT) model were used to calculate the specific surface area (SSA) and pore size distribution of the BACs. The thermal stability of the BAC was characterized using a thermal gravimetric analyzer (TGA, PerkinElmer, Waltham, MA, USA).

### 3.3. Adsorption Performances of Methylene Blue (MB)

The adsorption experiments were carried out in glass beakers, which were placed on a rotary shaker at 150 r/min at room temperature (25 ± 1 °C) for 120 min to achieve adsorption equilibrium. A quantity of 10 mg of BACs was introduced into 50 mL of 100 mg/L MB solution. Liquid samples were collected at different times (5, 10, 20, 30, 60, 120, 180, 240, and 360 min) to calculate adsorption kinetics. The adsorption kinetics were fitted through pseudo-first-order and pseudo-second-order models and intra-particle diffusion [50]. A quantity of 10 mg of BACs was added to an MB solution of 100–300 mL/L for 120 min at 25 °C of temperature. The equilibrium data of MB adsorption were calculated by isotherm models (Langmuir, Freundlich, Temkin, and Dubinin–Radushkevich) [29]. The effects of adsorption temperatures (15–45 °C) and solution pH (4–10) on MB removal were investigated. The pH value was controlled by the NaOH solution of 1 mol/L and HCl solution of 1 mol/L. Residual concentration of MB was determined using ultraviolet spectrophotometry (UV-Vis). A total of 2 mL of MB filtrates was added into a colorimetric tube and was determined at a wavelength of 665 nm. The adsorption capacity of MB was calculated by Equation (22):*q_e_* = (*C*_0_ − *C_e_*)*V/m*
(22)
where *q_e_* is the equilibrium adsorption capacity (mg/g); *C*_0_ and *Ce* are the initial and equilibrium concentration of MB in solution (mg/L), respectively; *V* is the volume of MB solution (mL); *m* is the mass of adsorbent used in the experiment (mg).

To investigate the regeneration capability of BACs, three cycles of the adsorption process were performed under optimum adsorption conditions. The BAC was regenerated by ethanol. The desorption experiment was carried out in a glass beaker, which was placed on a rotary shaker at 150 r/min at room temperature (25 ± 1 °C) for 180 min. Then the BAC was washed with deionized water. Finally, the BAC was dried at 105 °C until the mass stabilized for the next adsorption experiment.

## 4. Conclusions

A BAC with an ultrahigh specific surface was manufactured through co-pyrolysis of K_2_FeO_4_ and BC-450. At a temperature of 298 K, the optimum activation process of BAC was suggested to be at an activation temperature of 750 °C and activation time of 90 min, which showed the maximum adsorption capacity of 745.23 mg/g, calculated by the Langmuir model. The adsorption mechanism included pore filling, π–π conjugation, hydrogen bonding, and surface complexation with oxygen-containing functional groups. The BAC had an excellent adsorption capacity at a wide pH value. The Freundlich model was suitable to describe the isothermal adsorption of MB. The MB adsorption of the BAC was a multi-layer physisorption and chemisorption process. The adsorption process was endothermic. Temperature improved the adsorption property of the BAC. The removal rate of the BAC was 63.5% after three cycles. Therefore, it will be a promising purification agent to remove MB from wastewater based on its efficient and stable adsorption capacity.

## Figures and Tables

**Figure 1 molecules-28-03410-f001:**
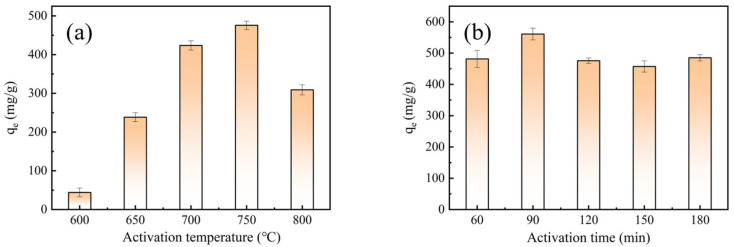
Adsorption capacities of the BACs in relation to (**a**) activation temperatures and (**b**) activation times.

**Figure 2 molecules-28-03410-f002:**
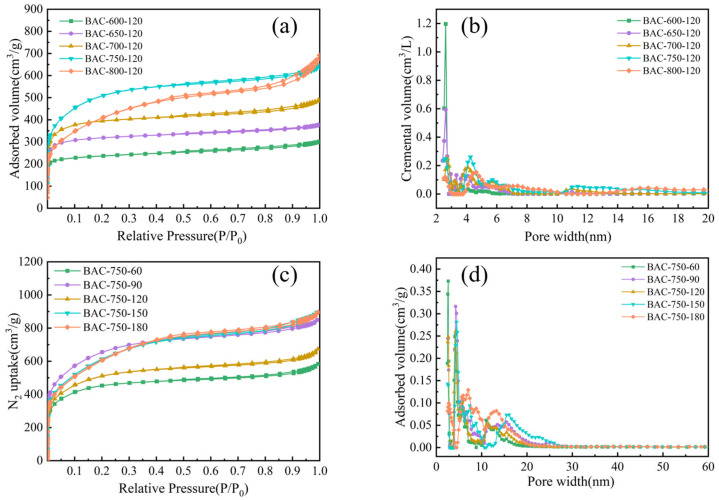
Pore characteristics of the BACs. (**a**,**c**) N_2_ adsorption/desorption isotherms; (**b**,**d**) pore-size distribution curves.

**Figure 3 molecules-28-03410-f003:**
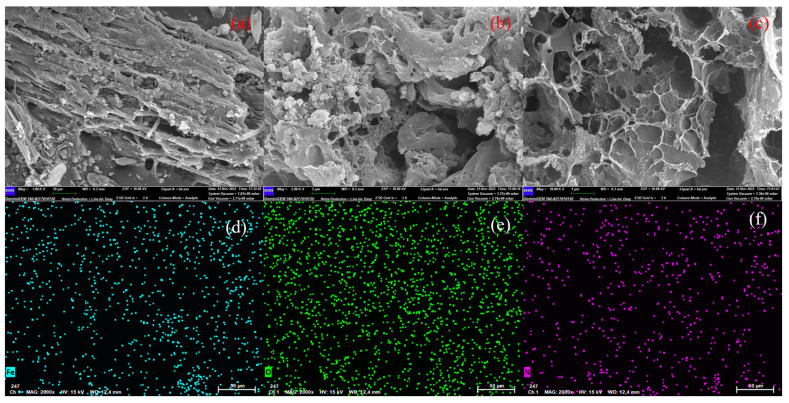
The morphology the BAC. (**a**–**c**) SEM images of BAC; (**d**) Fe; (**e**) O; (**f**) N.

**Figure 4 molecules-28-03410-f004:**
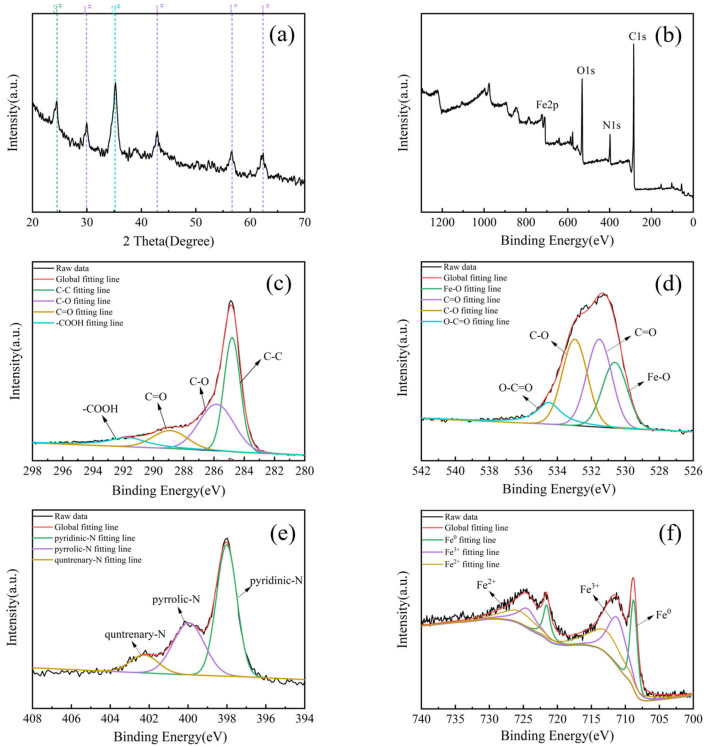
(**a**) XRD patterns of the BAC (1#-Fe_3_O_4_, 2#-Fe_2_O_3_, 3#-Fe_3_C); (**b**) XPS survey spectrum of the BAC; (**c**) C1s, (**d**) O1s, (**e**) N1s, and (**f**) Fe 2p.

**Figure 5 molecules-28-03410-f005:**
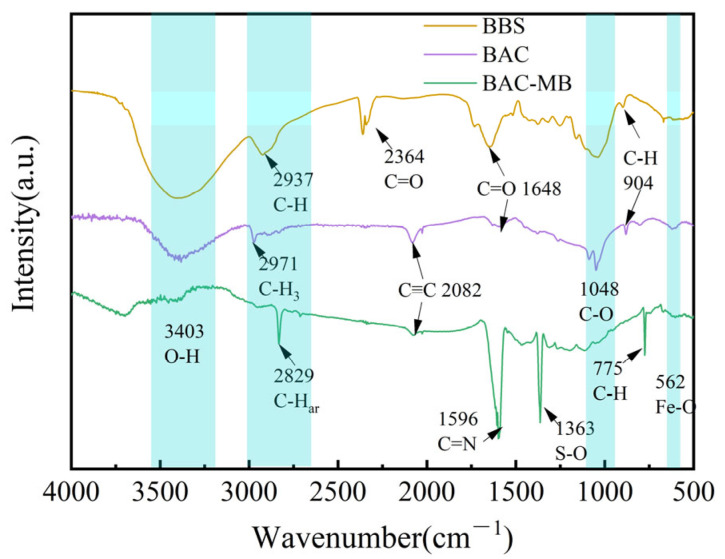
Surface functional groups of the BAC.

**Figure 6 molecules-28-03410-f006:**
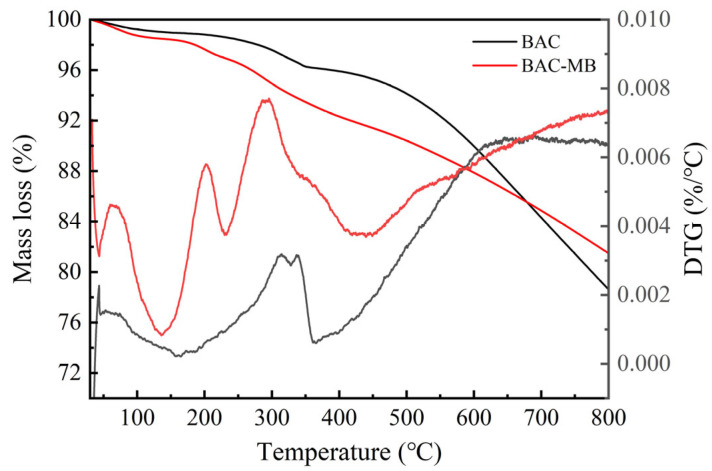
Thermal stability of the BAC.

**Figure 7 molecules-28-03410-f007:**
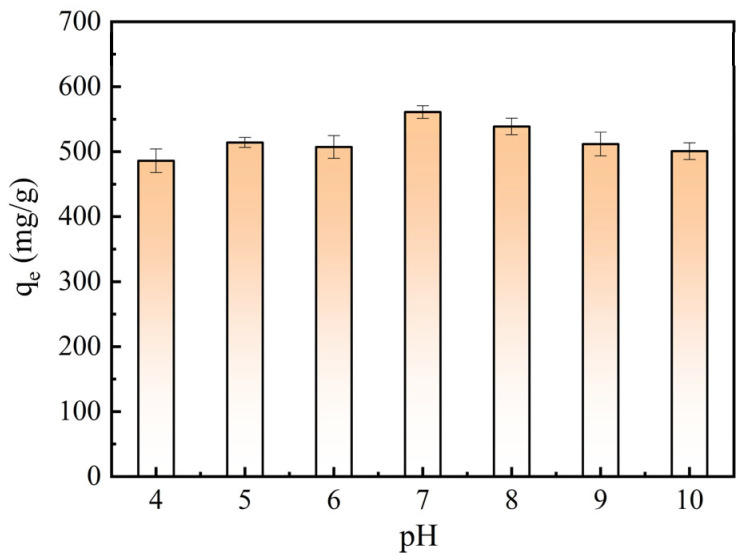
Effect of pH value on BAC adsorption properties.

**Figure 8 molecules-28-03410-f008:**
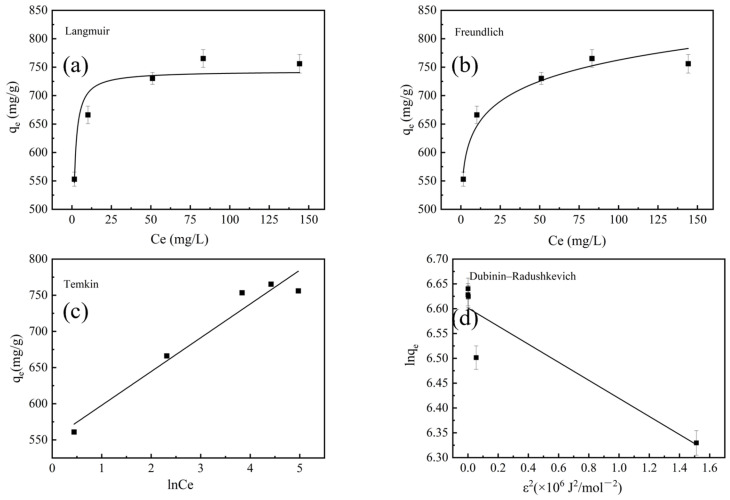
Fitting lines of the BAC at 298 K with Langmuir (**a**), Freundlich (**b**), Temkin isotherm models (**c**) and Dubinin–Radushkevich (**d**).

**Figure 9 molecules-28-03410-f009:**
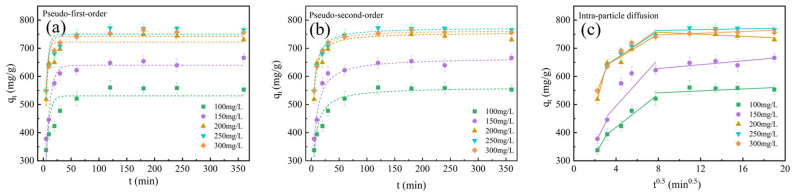
Plots of (**a**) pseudo-first-order, (**b**) pseudo-second-order, and (**c**) particle diffusion models of the BAC for MB.

**Figure 10 molecules-28-03410-f010:**
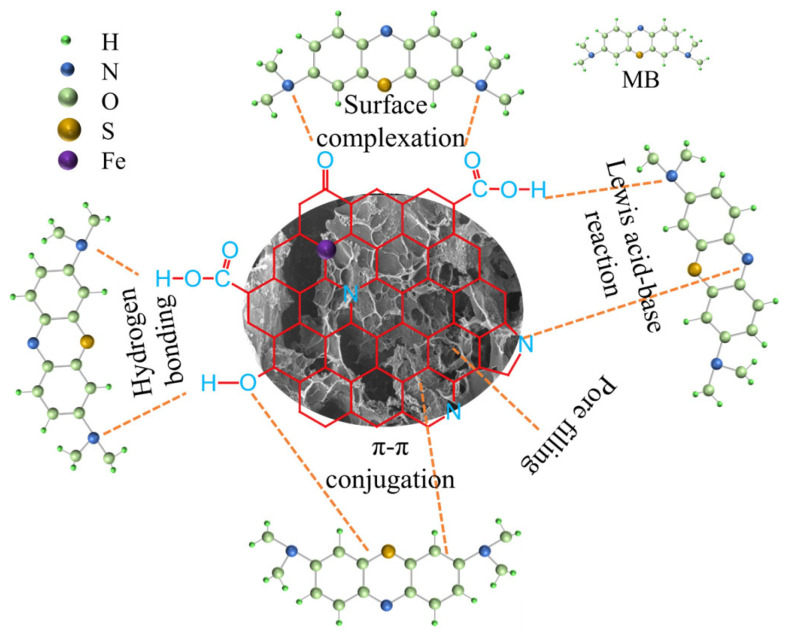
Illustration of adsorption modality between the BAC and MB.

**Figure 11 molecules-28-03410-f011:**
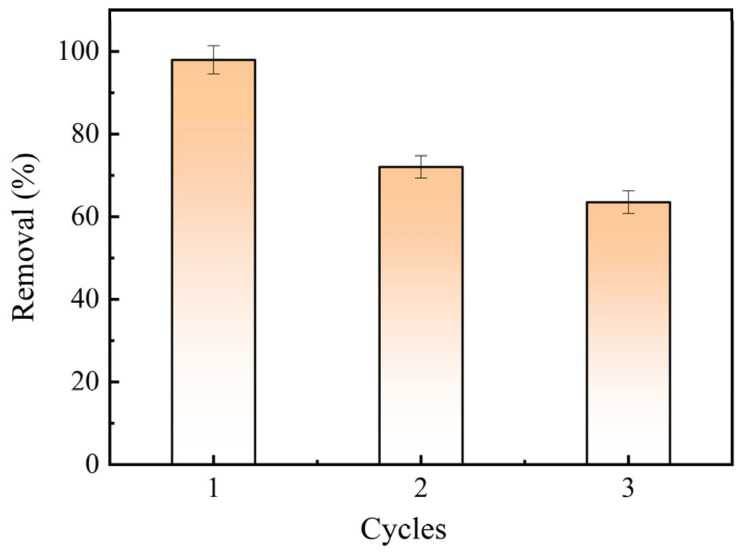
Reusability of the BAC.

**Table 1 molecules-28-03410-t001:** Surface areas and porosities of the BACs.

Samples	S_BET_ (cm^2^/g)	V_tot_ (cm^3^/g)	V_mic_ (cm^3^/g)	V_mic_/V_tot_ (%)
BAC-600-120	919.9	0.45	0.31	69.08
BAC-650-120	1243.0	0.59	0.44	73.95
BAC-700-120	1469.2	0.76	0.55	72.04
BAC-750-120	1835.1	1.04	0.67	64.23
BAC-800-120	1450.7	1.11	0.38	34.59
BAC-750-60	1649.0	0.90	0.62	68.14
BAC-750-90	2327.7	1.31	0.84	63.82
BAC-750-20	1835.1	1.04	0.67	64.23
BAC-750-150	2169.1	1.38	0.67	48.23
BAC-750-180	1729.0	0.87	0.65	74.62

**Table 2 molecules-28-03410-t002:** Isotherm parameters for adsorption of the BAC.

Isotherms	Parameters		
Langmuir	*Q_m_* (mg/g)	*K_L_* (L/mg) *R_L_*	R^2^
	745.23	1.79, 7.63–2.56 × 10^−3^	0.9414
Freundlich	*K_F_* ((mg/g) (L/mg)) 1/n	1/n	R^2^
	546.85	0.072	0.9664
Temkin	*K_T_*	*B*	R^2^
	135.26	46.683	0.9511
Dubinin–Radushkevich	*K* (mol^2^ kJ^2^)	*E* (kJ/mol)	R^2^
	0.1823	1.656	0.8454

**Table 3 molecules-28-03410-t003:** Kinetics parameters for adsorption of the BAC.

Kinetics	Parameters	*C*_0_ (mg/g)				
		100	150	200	250	300
	*q_e_,exp* (mg/g)	560.94	666.12	753.45	765.32	756.09
Pseudo-first-order kinetic model	*k*_1_ (min^−1^)	0.17	0.13	0.22	0.24	0.28
	*q_e_,cal* (mg/g)	530.89	639.49	742.94	750.18	721.86
	R^2^	0.7626	0.9223	0.8364	0.8661	0.9295
Pseudo-second-order kinetic-model	*k*_2_ (g/mg/min)	4.54 × 10^−4^	3.38 × 10^−4^	6.28 × 10^−4^	6.12 × 10^−4^	6.64 × 10^−4^
	*q_e_,cal* (mg/g)	561.79	666.67	756.18	773.87	764.38
	R^2^	0.9455	0.9937	0.9371	0.9902	0.9992
Intra-particle diffusion kinetic-model	*k*_1_ (g/mg/min^−1/2^)	61.063	73.342	139.330	96.851	93.682
	*C* _1_	201.362	213.897	207.245	332.194	339.125
	R_1_^2^	1	1	1	1	1
	*k*_2_ (g/mg/min^−1/2^)	37.526	35.017	22.961	23.143	21.950
	*C* _2_	268.519	380.975	565.232	572.379	582.382
	R_2_^2^	0.9854	0.7005	0.9265	0.9734	0.8674
	*k*_3_ (g/mg/min^−1/2^)	−0.656	3.111	−1.466	1.378	1.335
	*C* _3_	566.824	604.468	763.829	746.593	736.569
	R_3_^2^	0.7966	0.65626	0.51949	0.30059	0.41108

**Table 4 molecules-28-03410-t004:** Thermodynamic parameters for MB adsorbed on BAC.

ΔH (kJ/mol)	ΔS (kJ/mol/k)	Δ*G* (kJ/mol)			
		288 K	298 K	308 K	318 K
13.31	48.72	−6.28	−13.58	−17.10	−21.20

## Data Availability

Not applicable.

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
