# Peer review of "Activated Carbon with Ultrahigh Specific Surface Derived from Bamboo Shoot Shell through K2FeO4 Oxidative Pyrolysis for Adsorption of Methylene Blue"

_molecules, 2023, doi:10.3390/molecules28083410_

Round 1

Reviewer 1 Report

Review report on the manuscript titled:

Activated carbon with ultrahigh specific surface derived by 2 bamboo shoot shell through K2FeO4 oxidative pyrolysis for ad- 3 sorption of cationic dyes

Molecules 2023, 28, x. https://doi.org/10.3390/xxxxx

The manuscript describes the development of an activated carbon, with very high specific surface area derived from bamboo bark by oxidative pyrolysis of K2FeO4 for the adsorption of methylene blue. After review, the content of the manuscript is very interesting but requires serious improvement before being accepted for publication. My comments and remarks are as follows

Notes on the form:

The title should be improved to clarify the main points of the study and attract more interested readers. I suggest: “Development of a highly porous activated carbon from Bamboo shoot shells by oxidative pyrolysis of K2FeO4 for efficient removal of methylene blue”

1- English should be improved by a native speaker

2- Abstract should be improved and restructured in the following order: (1) problematic (2) objective of the study (3) method (5) main characterization results (6) adsorption results (maximum capacity, kinetics , thermodynamics and mechanism) (7) a small conclusion.

3- Key words: Replace the expression “adsorption property word” by “adsorption process”.

4- On line 25, you must write non-biodegradable instead of non-degradable.

5- You must write the complete word when it is cited for the first time in the text: abbreviation SSA line 31, AC line 37

6- Line 35: it is more correct to write: annual production and not annual yield.

7- Line 43 "The surface area, functional groups and metal species of the AC surface characteristic have a significant effect on the AC adsorption process"

This expression is badly worded and needs to be improved. Perhaps you wanted to write, "Inserting different elements into the surface of activated carbons, especially metals, can improve specific surface area and functional group richness".

8- Line 53 on the gap of this study "To our knowledge, there are no sufficient studies on the mechanism of adsorption of dyes" this sentence must be reformulated because the adsorption mechanism of dyes is widely studied.

9- The objective of the study is not well written, it is better to mention it in a separate paragraph at the end of the introduction "the objective of this study is to ..".

10- Normally the materials and methods section should be inserted after the introduction and before the results

11- It is necessary to add the characteristics of methylene blue: solubility, molar mass, UV/vis wavelength etc....

12- The sentence at line 264 needs to be improved

13- It is necessary to highlight the objective of each treated part before talking about the results.

14- In the graphs of Figure 1, it is necessary to add the error bars, specify the number of repetitions of the experiments and note the adsorption capacity as Qe (mg/g).

15- It is necessary to explain how the abbreviations of the activated carbons developed were adopted,

16- I think the authors confuse BBS and BSS to abbreviate the name of the used biomass "Bamboo shoot shells".

17- In figure 3, cm-1 instead of cm-1.

18- Add the values of the RL parameter in table 2.

19- The table of kinetic results is table 3 and not 2.

20- Intra-particle diffusion-Kineti mode in table 3 "kinetic" instead of "kineti".

21- k2(min-1) must be in g.mg-1.min-1 and not in min-1

Scientific remarks

22- In the materials and methods section, a lot of information is missing on the process of developing activated carbon.

23- It is necessary to add the HCl concentration of the carbon rinse and has your activated carbon been neutralized after the HCl rinse?

24- Several techniques are needed to study the characteristics of the developed activated carbon, including: EDS, TGA, TDA

25- Several informations on the adsorption processes carried out are missing: pH for the kinetic study, temperature, stirring speed, MB dosage method, etc.

26- In the material and method section, the authors cited only the carbonization at 450°C, while in the result section; they discussed the influence of the activation temperature on the adsorption capacity. I recommend detailing the protocol of the study in the material and method part so that readers can understand the study and be inspired by the manipulations by repeating them on other raw materials.

27- In the results part, it is necessary to explain why the adsorption capacity decreases at 800°C.

28- I recommend consulting the article below to improve the discussion on the evolution of adsorption capacity with increasing temperature for carbonization/activation https://doi.org/10.1016/j.jclepro .2023.136333

29- It should be explained why the adsorption capacity is optimal in 90 min and that it decreases after this residence time.

30- The widening of the pores with the increase in the activation temperature is not a function of the carbon oxidation phenomenon, but it is perhaps the result of the brutal and rapid release of the degraded material during the carbonization and creation with creation of large pores. At low temperature the releases are slow and not brutal, which ensures smaller pores. I recommend this article to improve your interpretations and discussions https://doi.org/10.1016/j.jclepro.2020.1256

31- It is necessary to specify the residence time in the study of the influence of the activation/carbonization temperature on the capacity and the porosity.

32- What do you mean by "the pore size is improved when the residence time increases from 60 to 180min"

33- How the authors determined the interactions established between MB molecules and the surface of the developed activated carbon, while only the infrared spectra of the biomass (BBS) and the developed activated carbon (BAC) are presented. The chemical mechanism must be treated and developed seriously in an independent part "mechanism" at the end of the manuscript before the conclusion.

34- Specify the time required to reach adsorption equilibrium

35- In figure 6.b, it appears that the quantity adsorbed at equilibrium is constant with the evolution of the adsorption temperature. The authors consider that the adsorption process is endothermic, the ideal is to calculate the enthalpy of adsorption by the van't hoff method to confirm whether the adsorption is endothermic or exothermic.

36- Line 176, the temperature of 30°C is insufficient for the desorption of molecules, it is necessary to specify on the section material and method, the exact technique to carry out these experiments.

37- Line 183-184 the validation of the Langmuir model gives several indications, this article can help you improve your interpretations.... https://doi.org/10.1016/j.biortech.2022.128162

38- Line 185 you want to write the maximum adsorption capacity?

39- It is recommended to calculate the initial kinetics using the relationship

h(mg.g-1.min-1)=k2*qe²

40- It is recommended to calculate the thermodynamic parameters (van't hoff) of adsorption (enthalpy, entropy and free enthalpy) to better understand the mechanism of adsorption.

41- The mechanism must be confirmed by an FT-IR analysis before and after the adsorption and not by the literature.

42- Is the activated carbon developed generable and reusable?

43- In the results part, I recommend starting with the characterization of the materials, then the adsorption processes.

44- More explanation and information on the concept of oxidative pyrolysis and especially the intervention of the oxidizing agent K2FeO4

45- The conclusion needs serious improvement.

Reviewer 2 Report

The manuscript has innovation and extensive characterization. However, some corrections must be made for it to be recommended for publication.

Since only one dye was analyzed, I advise that "cationic dyes" be replaced by "methylene blue" in the title.

Please explain this statement further. (lines 49-50);

Please substitute the word "optimum" there is no statistical basis for this statement (abstract and section 2.1);

Authors should enter the standard deviation in Figure 1a,b;

In table 1, "BAC- X m " should contain the pyrolysis temperature;

Please rewrite this sentence as it is confusing. Lines 114-116;

Figure 3: Please identify the peak at around 3000 cm-1 for BAC; identify peaks between 2500-2000 cm -1 for BBS; identify the peak between 1000 and 750 cm-1 for BBS and BAC;

Please enter the standard deviation in figures 6-8;

Linear isotherm models must be replaced by quadratic models; several easily found studies discourage the use of linear models. If the authors insist on keeping the linear models, this discussion must be deepened in the manuscript;

Please correct the word "Patameters" in table 2;

the pseudo second-order linear model should be substituted for the quadratic;

Please insert the geographical coordinates where the samples were collected; (section 3.1)

Section 3.2: How was activated carbon neutralized?

Please explain better the meaning of "residence time";

Insert the HCl concentration in section 3.2;
The yield of the samples must be mentioned in the manuscript, due to the proposal of the manuscript;

The desorption analyses are also recommended.

Round 2

Reviewer 1 Report

The revision is correct and the manuscript has been improved except that the authors should replace the term BBS by BSS throughout the manuscript.

Reviewer 2 Report

All comments have been responded to point-by-point by the author, and I suggest that the manuscript be accepted.